

# Ecological engineering in low land rice for brown plant hopper, *Nilaparvata lugens* (Stål) management

Yogesh Yele[1], Subhash Chander[2], Sachin S. Suroshe[3], Suresh Nebapure[3], Prabhulinga Tenguri[4] and Arya Pattathanam Sundaran[3]

[1] National Institute of Biotic Stress Management, Raipur, Chattisgarh, India
[2] National Centre for Integrated Pest Management, New Delhi, Delhi, India
[3] Indian Agricultural Research Institute, New Delhi, Delhi, India
[4] Central Institute for Cotton Research, Nagpur, Maharashtra, India

Corresponding authors
Yogesh Yele, yogeshyele13@gmail.com
Sachin S. Suroshe, sachinsuroshe@gmail.com

## ABSTRACT

Rice field bunds and edges can act as near crop habitats, available for planting flowering plants to attract and conserve the natural enemies. We evaluated the effect of ecological engineering on the incidence of Brown Planthopper (BPH), *Nilaparvata lugens* (Stål) (Hemiptera; Delphacidae) and the abundance of its predators in the rice variety Pusa Basmati-1121. Plots included the oilseed crops *viz.* sesamum, sunflower and soybean, with plantings of flowering crops marigold, balsam and gaillardia as bund flora around the edges of rice plots. Ecologically engineered plots contained both crops+flowers and resulted in a significantly reduced BPH population per hill in rice plots for 2019 (6.3) and 2020 (9.4) compared to the control plots (9.8 and 14.4). Ecologically engineered plots also witnessed the delayed appearance of BPH during each growing season. Peak BPH populations are lower in the ecologically engineered plots than in the control grounds. Furthermore, the activity of natural enemies, *viz.,* spiders, mirid bugs and rove beetles was the highest in rice fields planted with oilseed crops like sesamum, sunflower and soybean. Olfactory response studies showed that the attraction response of spiders toward sesamum and balsam leaves was more significant than in other crop plants. Rice yield was enhanced in plots planted with crops+flowers during both seasons compared to control plots. Planting of oilseed crops plants such as sesamum, sunflower and soybean with flowering crops such as marigold, balsam and gaillardia as bund flora on the bunds around the main rice field enhanced the natural enemy activity, suppressed the planthopper population, and increased yields. Based on the results, we recommend including ecological engineering techniques as one of the management components in the Integrated Pest Management programme for rice crops.

## INTRODUCTION

Rice, *Oryza sativa* L. is the world's most crucial staple food worldwide, providing nutrition to more than half of the world's population. Many biotic and abiotic stresses act as the

bottlenecks challenging rice production. Extensive rice cultivation systems, especially monoculture, have increased problems related to rice cultivation, including insect pests, diseases, and weeds (*Behura, Sen & Kar, 2011*). Several decades of agricultural intensification and agrochemical overuse have led to depletion of natural enemy populations (*Matsumura et al., 2008*). In the absence of sufficient natural enemies, pesticide-survived pests maintain that inoculum population during the off-season. This will then outbreak during the subsequent season infestation (*Yele et al., 2020*). Moreover, survivors sustain insecticide resistance and invasive pest population infestations over rice varieties (*Horgan et al., 2015*). Indiscriminate insecticide use by rice farmers causes the resurgence of pest outbreaks, including planthoppers outbreaks in several Asian regions (*Catindig et al., 2009*; *Cheng, 2009*). Ecological pest outbreaks are associated with the reduced diversity and efficiency of the natural enemies in the rice crop ecosystem.

Ecological engineering involves deliberately manipulating the habitat for the benefit of society and the natural environment (*Horgan et al., 2016*). Ecological engineering for pest management mainly aims at increasing the abundance, diversity, and function of natural enemies in agricultural habitats by providing them with refuge and supplementary food resources (*Gurr, 2009*; *Lv et al., 2015*). Application of this method has resulted in successful cases in crop production systems, thereby solving pest management problems. Planting buckwheat, *Fagopyrum esculentum* Moench, as a cover crop in vineyard and alyssum, *Lobularia maritima* (L.) Desv., between the rows of vegetables provides pollen and nectar to predators and parasitoids, which results in enhanced biological control (*Berndt, Wratten & Scarratt, 2006*; *Gillespie, Rogers & Ainsworth, 2011*). Flowering plants and weeds provide many resources for the survival of natural enemies such as alternative prey/hosts, pollen, nectar, and microhabitats. The concept of pest management through ecological engineering is still in its infancy in the rice ecosystem in India. Studies on ecological engineering for rice pest management have primarily focused on integrating flower and/or vegetable strips into rice landscapes.

Planting rice bunds with okra, mung beans, and string beans increased the structural diversity of predators in the rice fields. Consequently, spider abundance increased, and the plant hopper to spider ratio was lower among rice plants in fields close to planted bunds (*Horgan et al., 2016*). *Zhu et al. (2014)* studied the influence of various plant species on the performance of the predatory mirid bug, *Cyrtorhinus lividipennis* Reuter, a key natural enemy of rice planthoppers. The presence of flowering plants, such as *Tagetes erecta* L., *Tridax procumbens* L., *Emilia sonchifolia* (L.), and *Sesamum indicum* L., around the rice plots increased the abundance and survival of *C. lividipennis*. The predation efficiency and consumption of *Nilaparvata lugens* by *C. lividipennis* increased plots planted with flowers. Among flowering plants, *S. indicum* was favourable and strongly promoted host predation by *C. lividipennis*. These studies have suggested that *S. indicum* is the best suited floral component for ecological engineering in rice. *Chandrasekar, Muthukrishnan & Soundararajan (2017)* recommended using weed strips of *Echinochloa colona* (L.) and *E. crusgalli* in the rice ecosystem to increase the availability of mirid bugs. *Zheng et al. (2017)* used the banker plant system in rice for the biological control of BPH, *N. lugens*, and plant hopper, *N. muiri*. In the banker plant system, a grass species, *Leersia sayanuka*

Ohwi, was planted adjacent to rice fields. *Leersia sayanuka* is a host plant for *N. muiri,* but it could not complete the life cycle on rice.

Similarly, BPH could not complete the life cycle on grass, *L. sayanuka.* The egg parasitoid *Anagrus nilaparvatae* (Pang et Wang) actively parasitizes eggs of both BPH and *N. muiri.* The *L. sayanuka* improved the establishment and persistence of the egg parasitoid, *A. nilaparvatae.* Moreover, BPH densities were significantly lower in rice fields with a banker plant system than in control rice fields. *Jado et al. (2019)* recently demonstrated improvements in the biological control potential of parasitoids on aphids through exposure to flowering plants. Long-term exposure to buckwheat (*F. esculentum*), alyssum (*L. maritima*), and white rocket (*Diplotaxis erucoides* L.) flowers greatly enhanced the longevity, the potential fecundity, and parasitism rate of *Aphidius colemani* Vieron on aphid *Myzus persicae* (Sulzer).

Conservation of biodiversity and optimization of ecosystem functions are urgently required for sustainable agriculture. Ecological management methods are an efficient means of achieving these goals, while, at the same time, restoring the ecology of rice landscapes is also necessary (*Horgan et al., 2016*). Ecological engineering techniques have considerable potential in rice pest management, including BPH, for reducing pesticide dependence and slowing the breakdown of varietal resistance. Identifying flowering plants that selectively favors natural enemies over insect pests is a crucial consideration for ecological engineering. However, little information is available on the optimal fauna and flora species to be used for this cause. Selecting appropriate flowering plants for the attraction and enhanced biological activity and conservation of natural enemies is essential for the successful ecological engineering. Studies shall start evaluating the effect of ecological engineering in rice on BPH incidence of and the abundance of its natural enemies.

## MATERIALS AND METHODS

### Field preparation and transplanting

Experiments involving ecological engineering studies on BPH and their natural enemy population were conducted in the rice fields by using the rice cultivar *Pusa Basmati 1121* during kharif (rainy season) in 2019 and 2020. A tractor equipped with a drawn cultivator and rotavator was used to plough the main field twice to obtain fine tilth. All weeds and previous crop stubbles were removed from the field, submerged in water for 2–3 days and puddled 2–3 times, followed by leveling. Plots were 5× 4 m in size with ridges on all sides, spaced 1 m apart. Transplanting was performed on July 22, 2019 and on July 30, 2020. Two seedlings were planted per hill, spaced at 15× 20 cm. All plots were surrounded by ridges, filling the gaps after a week to ensure a uniform plant population in each plot.

### Experimental treatments and layout

This experiment was performed to study the effect of field crops and flowering crops surrounding the rice fields on the abundance of BPH and their natural enemies. Three oilseed crops, namely sesamum (*Sesamum indicum* L.), sunflower (*Helianthus annuus* L.), and soybean (*Glycine max* L.), and three flowering crops, namely marigold (*Tagetes*

*erecta* L.), balsam (*Impatiens balsamina* L.), and gaillardia (*Gaillardia pulchella* Foug.), were selected for the study. The study focused on the interaction between oilseed and flowering crops and evaluated the effect of natural weeds on the abundance of pests and their natural enemy populations in rice crops. To conduct these studies, we designed the treatments as T1 = oilseed crops; T2 = flowering crops; T3 = natural weeds (No weeding); T4 = oilseed crops + flower crops; T5 = control rice plots with all recommended agronomic practices.

On the bunds adjacent to respective treatments, sesamum, sunflower and soybean seeds were directly sown by dibbling Marigold, balsam, and gaillardia plants were first raised in the nursery, and, at an appropriate time, transplanted to rice bunds adjacent to the proper treatments. The oilseed crops and flowering plants were also grown in plastic plots and placed around rice plots in respective treatments at the appropriate time. Placing oilseed and flowerings in bunds and staggering maintained a more extended flowering in the plot. A 1-meter channel between replications allowed managing the experiment in completely randomized block design (CRBD) with five treatments and four replications.

## Observations and statistical analysis

Ten random hills in each plot presented the incidence of BPH and its natural enemies, spider (*Lycosa pseudoannulata* (Bosenberg and Strand)), mirid bug (*Cyrtorhinus lividipennis* Reuter), and rove beetles (*Paederus fuscipes* Curtis). The observation interval was 10 days untill the crop was harvested. Records per hill include the total BPH population and natural enemies such as spider, mirid bug, and rove beetle. Yield data were recorded after harvesting and expressed as tons per hectare. Thus, data obtained for BPH and natural enemies subjected to a two-way analysis of variance (two-way ANOVA), and the significance of differences between the treatments and weeks was tested using $F$-tests. By contrast, the treatment means compared least significant differences (LSD) at $P = 0.05$. Rice yield data passed ANOVA and means comparison by LSD at $P = 0.05$.

## Olfactory response studies

A Y-tube olfactometer (*Fand et al., 2020*) enables the evaluation of olfactory attraction responses of spider to the odors of flowering plants. The experimental arena consisted of a Y-tube having two arms with the total length of 15 cm, with each arm being 7.5 cm long and having a 1- cm internal diameter. One arm of the Y-tube was attached to a plant odor source, and another arm to the clean air source. A vacuum pump and a flow meter present at the base end maintained a constant air inflow from both arms. Teflon tubing sections of an intermediate diameter were used as tight-fitting unions to plant source and vacuum lines. A nylon mesh barrier between the teflon union and the glass T-tube prevented the spiders from crawling to the tube ends. The complete Y-tube assembly was stationed on a foam platform to minimize vacuum pump induced ambient vibrations.

Fresh plant leaves were collected from the field in an airtight zipper plastic bag. Spiders were collected from the rice fields, individually stored in glass vials, and starved for 2 h before participating in the attraction response experiments. One arm of the Y- tube was connected by teflon tubes to the plastic bags enclosing plant leaves. Another arm was a source of charcoal passed clean air. A single starved spider was released at the base of the

Y-tube and observed for 20 min, allowing it to choose between the two arms. Most spiders began moving towards the arms within a few minutes of release. A new spider replaced a non-active individual. The active spiders moving towards the arms and touching the mesh barrier were respondents, whereas the others were non-respondents. Twenty spiders per treatment were used for the bioassay. Finally, the percent attraction and a two-tailed paired $t$-test (0.01 and 0.05, respectively) indicated the significant difference between the means of the percent spiders attracted towards the plant source.

## RESULTS

### Effect of ecological engineering on BPH incidence

The BPH population (nymph, female, and male) differed significantly across the treatments ($F = 4.32$, $P = 0.002$ and $F = 11.5$, $P < 0.001$) and weeks ($F = 81.2$, $P < 0.001$) during kharif 2019 and 2020. The BPH population first appeared at 36 Standard Meteorological Week (SMW) during kharif 2019 and 2020 (Tables 1 & 2). The BPH per hill population showed no significantly difference until 64 days after transplanting (DAT) across the treatments in the kharif seasons. During kharif 2019, all treatments experienced the peak BPH population at 94 DAT (43 SMW). The mean BPH population density was the highest in the control plots ($9.8 \pm 3.9$ BPH/hill) and did not differ significantly from that in the plots treated with natural weeds ($11.5 \pm 4.2$ BPH/hill) (Table 1).

The BPH population density was significantly lower in treatments with crops ($5.2 \pm 1.8$ BPH/hill) than in other treatments and the control (Fig. 1). The highest peak BPH population was recorded in the control ($27.8 \pm 9.5$ BPH/hill) at 94 DAT, whereas the lowest peak BPH population existed in treatments with crop plants ($15.1 \pm 3.9$ BPH/hill) at 94 DAT (Fig. 2). The peak population was significantly lower in the treatments with flowers and crops+flowers than in the treatment with natural weeds and the control. The mean population density was also lower in the treatment than in the control. During kharif 2020, the peak population appeared at 82 DAT (42 SMW) in all treatments, except for treatments with crops, in which the peak population appeared at 71 DAT (41 SMW) (Table 2). The lowest BPH mean population density was observed in the crops+flowers treatment ($9.4 \pm 3.6$ BPH/hill), whereas the highest population was recorded in the control treatment ($14.4 \pm 5.1$ BPH/hill) (Fig. 1). After the crops+flowers treatment, the BPH population ranged from $0.1 \pm 0.1$ BPH/hill at 41 DAT to $23.6 \pm 0.4$ BPH/hill at 71 DAT, which was the lowest population among all the treatments (Table 2). During kharif 2020, the highest peak population was observed with the control treatment ($33.7 \pm 2.1$ BPH/hill). The lowest peak population for the crops+flowers treatment ($23.6 \pm 0.4$ BPH/hill) occurred at 71 DAT (41 SMW) (Fig. 2). Rice plots surrounded with crops and flowers harbored a significantly lower BPH population than those treated with natural weeds and control. Overall, the BPH population was higher during kharif 2020 than during kharif 2019, irrespective of the treatments. Control plots harbored a significantly higher BPH population than the plots subjected to other treatments. By contrast, rice plots with oilseed crop plants, flowering plants, and a combination of crops+flowers exhibited lower BPH populations than the weedy and control plots.

Yele et al. (2023), *PeerJ*, DOI 10.7717/peerj.15531
**Table 1  Incidence of BPH population in ecologically engineered rice fields during kharif 2019.**

| Treatments | BPH population/hill[*] | | | | | | | | |
|---|---|---|---|---|---|---|---|---|---|
| | 44 DAT 36SMW | 54 DAT 37 SMW | 64 DAT 39 SMW | 74 DAT 40 SMW | 84 DAT 42 SMW | 94 DAT 43 SMW | 104 DAT 45 SMW | 114 DAT 46 SMW | Mean ±SE |
| **T1** Crops (sesamum+ sunflower+ soybean) | 0.08 ± 0.1 | 0.8 ± 0.1 | 1.1 ± 0.3 | 5.7 ± 1.3 | 10.9 ± 2.6 | 15.1 ± 3.9 | 4.6 ± 0.6 | 3.1 ± 0.5 | 5.2 ± 1.8 |
| | (1.0 ± 0.03) | (1.3 ± 0.04) | (1.4 ± 0.1) | (2.5 ± 0.2) | (3.4 ± 0.3) | (3.9 ± 0.4) | (2.3 ± 0.1) | (2.0 ± 0.1) | (2.2 ± 0.3)[c] |
| **T2** Flowers (marigold+balsam+ gaillardia) | 0.2 ± 0.1 | 0.7 ± 0.04 | 1.4 ± 0.4 | 5.6 ± 1.0 | 13.0 ± 4.9 | 19.9 ± 6.2 | 4.4 ± 0.4 | 3.4 ± 0.4 | 6.1 ± 2.4 |
| | (1.1 ± 0.04) | (1.3 ± 0.01) | (1.5 ± 0.1) | (2.5 ± 0.2) | (3.5 ± 0.6) | (4.4 ± 0.6) | (2.3 ± 0.1) | (2.1 ± 0.1) | (2.3 ± 0.4)[b] |
| **T3** Natural weeds | 0.05 ± 0.1 | 0.9 ± 0.1 | 1.2 ± 0.2 | 7.7 ± 1.8 | 17.0 ± 5.0 | 22.0 ± 5.4 | 7.1 ± 0.3 | 4.8 ± 0.3 | 7.6 ± 2.8 |
| | (1.0 ± 0.0) | (1.3 ± 0.06) | (1.4 ± 0.1) | (2.9 ± 0.3) | (4.1 ± 0.5) | (4.6 ± 0.5) | (2.8 ± 0.1) | (2.4 ± 0.1) | (2.6 ± 0.4)[a] |
| **T4** Crops+Flowers | 0.07 ± 0.1 | 0.8 ± 0.2 | 1.3 ± 0.1 | 7.1 ± 1.3 | 10.9 ± 1.5 | 21.5 ± 5.6 | 5.6 ± 0.2 | 3.5 ± 0.3 | 6.3 ± 2.5 |
| | (1.0 ± 0.03) | (1.3 ± 0.07) | (1.5 ± 0.06) | (2.8 ± 0.2) | (3.4 ± 0.2) | (4.6 ± 0.6) | (2.5 ± 0.05) | (2.1 ± 0.1) | (2.4 ± 0.4)[ab] |
| **T5** Control | 0.1 ± 0.1 | 0.7 ± 0.1 | 1.9 ± 0.2 | 10.4 ± 3.0 | 26.6 ± 4. | 27.8 ± 9.5 | 6.9 ± 0.5 | 3.7 ± 0.5 | 9.8 ± 3.9 |
| | (1.1 ± 0.03) | (1.3 ± 0.06) | (1.7 ± 0.1) | (3.2 ± 0.4) | (5.2 ± 0.4) | (5.1 ± 0.8) | (2.8 ± 0.1) | (2.1 ± 0.1) | (2.8 ± 0.5)[a] |
| Mean ±SE | 0.1 ± 0.03 | 0.8 ± 0.03 | 1.4 ± 0.1 | 7.3 ± 0.8 | 15.7 ± 2.9 | 21.3 ± 2.0 | 5.7 ± 0.5 | 3.7 ± 0.2 | |
| | (1.0 ± 0.0)[f] | (1.3 ± 0.01)[ef] | (1.5 ± 0.04)[e] | (2.8 ± 0.1)[c] | (3.9 ± 0.3)[b] | (4.5 ± 0.2)[a] | (2.5 ± 0.1)[c] | (2.1 ± 0.06)[cd] | |

**Notes.**

Treatments, F = (4.32), LSD = (0.3), $P = 0.002$.

Weeks, F = (81.2), LSD = (0.38), $P < 0.001$.

Interactions, F = (0.7), LSD = (NA), $P = 0.753$.

Planthopper count with different superscript differ significantly.

[*] Average of ten replications.

Numbers in parenthesis are SQRT (X+1) values, SMW-Standard Meteorological Wek, DAT-Days After Transplanting.

**Table 2  Incidence BPH population in ecologically engineered rice fields during kharif 2020.**

| | BPH population/hill[*] | | | | | | | |
|---|---|---|---|---|---|---|---|---|
| **Treatments** | **41 DAT**<br>**36 SMW** | **51 DAT**<br>**38 SMW** | **61 DAT**<br>**39 SMW** | **71 DAT**<br>**41 SMW** | **82 DAT**<br>**42 SMW** | **91 DAT**<br>**44 SMW** | **101 DAT**<br>**45 SMW** | **Mean ±SE** |
| **T1** Crops (sesamum+ sun- flower+ soybean) | 0.2 ± 0.1 | 2.5 ± 0.7 | 5.1 ± 0.5 | 24.2 ± 3.9 | 23.9 ± 1.3 | 15.5 ± 1.4 | 2.4 ± 0.0 | 10.5 ± 4.0 |
| | (1.1 ± 0.10) | (1.8 ± 0.2) | (2.5 ± 0.1) | (5.0 ± 0.4) | (5.0 ± 0.1) | (4.0 ± 0.2) | (1.8 ± 0.0) | (3.0 ± 0.6)[bc] |
| **T2** Flowers (marigold+balsam+ gaillardia) | 0.1 ± 0.1 | 2.4 ± 0.2 | 4.1 ± 0.3 | 22.8 ± 2.7 | 23.0 ± 1.9 | 14.8 ± 3.1 | 2.3 ± 0.4 | 9.9 ± 3.8 |
| | (1.0 ± 0.0) | (1.8 ± 0.0) | (2.3 ± 0.1) | (4.9 ± 0.3) | (4.9 ± 0.2) | (3.9 ± 0.4) | (1.8 ± 0.1) | (2.9 ± 0.6)[c] |
| **T3** Natural weeds | 0.0 ± 0.0 | 3.2 ± 0.4 | 4.9 ± 0.2 | 21.5 ± 3.8 | 29.3 ± 3.7 | 17.7 ± 1.3 | 3.9 ± 1.0 | 11.5 ± 4.2 |
| | (1.0 ± 0.0) | (2.0 ± 0.1) | (2.4 ± 0.0) | (4.7 ± 0.4) | (5.5 ± 0.3) | (4.3 ± 0.1) | (2.2 ± 0.2) | (3.2 ± 0.6)[b] |
| **T4** Crops+Flowers | 0.1 ± 0.1 | 2.1 ± 0.5 | 4.6 ± 0.3 | 19.6 ± 2.6 | 23.6 ± 0.4 | 14.0 ± 1.0 | 0.1 ± 0.1 | 9.4 ± 3.6 |
| | (1.0 ± 0.0) | (1.7 ± 0.1) | (2.4 ± 0.1) | (4.5 ± 0.3) | (5.0 ± 0.0) | (3.9 ± 0.1) | (1.8 ± 0.0) | (2.9 ± 0.6)[c] |
| **T5** Control | 0.1 ± 0.1 | 3.8 ± 0.6 | 6.5 ± 0.4 | 29.2 ± 5.0 | 33.7 ± 2.1 | 21.4 ± 0.9 | 6.4 ± 1.2 | 14.4 ± 5.1 |
| | (1.0 ± 0.0) | (2.2 ± 0.1) | (2.7 ± 0.1) | (5.4 ± 0.5) | (5.9 ± 0.2) | (4.7 ± 0.1) | (2.7 ± 0.2) | (3.5 ± 0.7)[a] |
| Mean ±SE | 0.1 ± 0.0 | 2.8 ± 0.3 | 5.1 ± 0.4 | 23.4 ± 1.6 | 26.7 ± 2.1 | 16.7 ± 1.3 | 3.4 ± 0.8 | |
| | (1.0 ± 0.0)[g] | (1.9 ± 0.1)[f] | (2.5 ± 0.1)[d] | (4.9 ± 0.2)[b] | (5.2 ± 0.2)[a] | (4.2 ± 0.2)[c] | (2.1 ± 0.2)[e] | |

**Notes.**

Treatments, F = (11.5), LSD = (0.2), $P < 0.001$.

Weeks, F = (345.2), LSD = (0.24), $P < 0.001$.

Interactions, F = (0.8), LSD = (NA), $P = 0.68$.

Planthopper count with different superscript differ significantly.

[*]Average of ten replications.

Numbers in parenthesis are SQRT (X+1) values, SMW-Standard Meteorological Wek, DAT-Days After Transplanting.

## Effect of ecological engineering on the abundance of natural enemies

For successive kharif seasons in 2019 and 2020, the spider *L. pseudoannulata* population was monitored at 40 DAT in the rice plots. The populations differed significantly between the treatments ($F = 14.7$, $P < 0.001$ and $F = 47.9$, $P < 0.001$) and weeks ($F = 29.6$, $P < 0.001$ and $F = 13.8$, $P < 0.001$) in both kharif seasons (Tables 3 & 4). In general, no difference was observed in the abundance of spider populations during kharif 2019 and 2020. During kharif 2019, the rice plots surrounded with crops+flowers experienced the highest abundance of the spider population ($2.5 \pm 0.3$ spider/hill), which ranged from $1.3 \pm 0.2$ spider/hill at 44 DAT to $3.9 \pm 0.3$ spider/hill at 104 DAT (Table 3). On the other hand, the lowest spider populations were observed in the natural weed rice plots ($1.7 \pm 0.2$ spider/hill). The peak spider population during kharif 2019 was observed at during 104 DAT in all the treatments, while the peak occurred in the crops+flowers treatment ($3.9 \pm 0.3$ spider/hill) (Table 3). Treatments with crops and flowers alone also led to a significantly higher spider population than the control treatment. During kharif 2020, the spider population was higher in the crop ($2.4 \pm 0.2$ spider/hill) and crop+flower ($2.3 \pm 0.2$ spider/hill) treatments than in the other treatments and control plots (Table 4). The peak spider population significantly differed between the treatments, and the highest spider population was observed after the crops treatment ($3.1 \pm 0.3$ spider/hill) at 82 DAT, whereas the lowest spider population was observed after the control treatment ($1.1 \pm 0.1$ spider/hill), which was at par with the population observed after the natural weed treatment ($1.3 \pm 0.1$ spider/hill). In general, spider abundance was higher in rice plots planted with

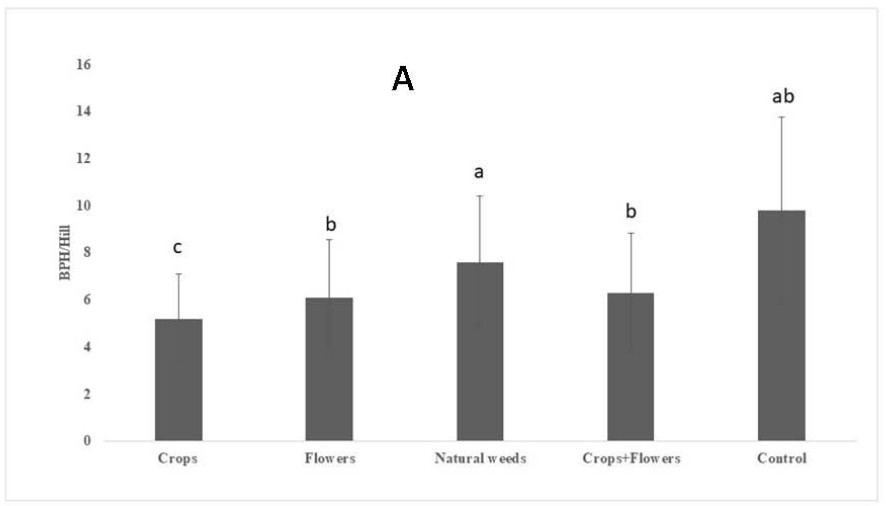

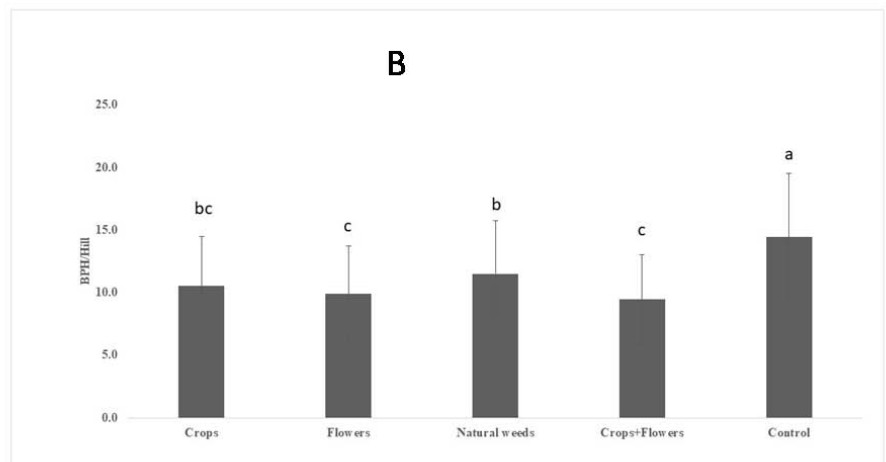

Figure 1 Mean BPH population in ecologically engineered rice fields during kharif 2019 (A) and 2020 (B).

crops+flowers, crops, and flowers than in the control plots. Crops and flower diversity along the rice plots enhanced the spider abundance in these plots (Table 4).

The mirid bug (*C. lividipennis*) population significantly differed across the treatments and weeks in both seasons (Tables 5 & 6). The abundance of the mirid bug was higher in kharif 2019 than in kharif 2020. During kharif 2019, the mirid bug population was present on all the observation weeks, while during kharif 2020, it first appeared in the field at 82 DAT. A significantly higher mirid bug population was after the treatments with crops (3.3 ± 1.9 mirid/hill), flowers (3.3 ± 2.0 mirid/hill), and crops+flowers (2.8 ± 1.7 mirid/hill), than after the control treatment (1.8 ± 1.1 mirid/hill) during kharif 2019 (Table 5). After the natural weeds treatment, the mirid bug population was the lowest (1.5 ± 0.8 mirid/hill) but was at par with the population after control treatment. The highest peak mirid bug population was observed after the crops treatment (12.5 ± 1.0
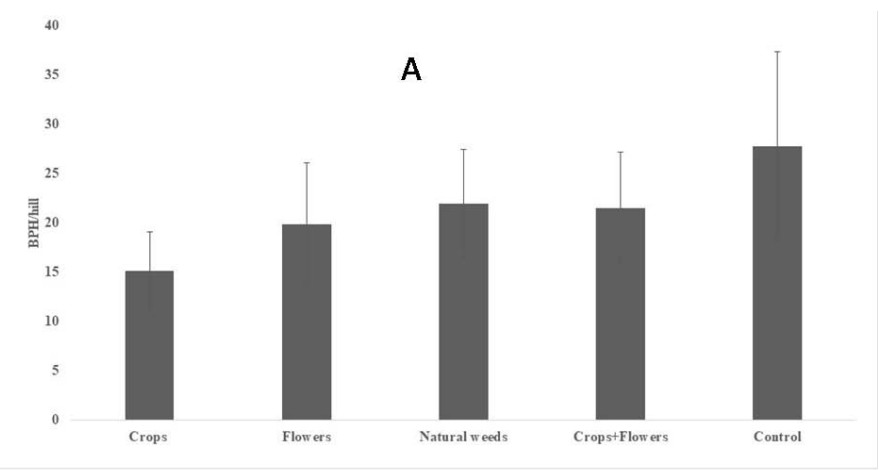

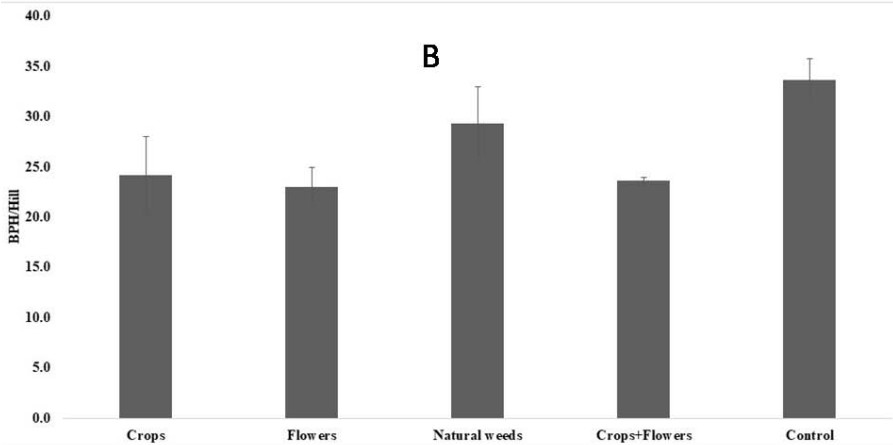

**Figure 2** Peak BPH population in ecologically engineered rice fields during kharif 2019 (A) and 2020 (B).

mirid/hill) at 104 DAT, followed by the flowers treatment (12.4 ± 0.6 mirid/hill) at 114 DAT and the crops+flowers treatment (10.9 ± 0.6 mirid/hill) at 104 DAT (Table 5). Until 94 DAT, the mirid population was lower but stable. However, the population significantly increased after 100 DAT irrespective of the treatment, which coincided with the higher BPH population on the rice plots. Similar to that in kharif 2019, the mirid bug population was higher after the treatments with crops (1.0 ± 0.3 mirid/hill), flowers (1.1 ± 0.2 mirid/hill), and crops+flowers (1.4 ± 0.4 mirid/hill) than after the control treatment (0.5 ± 0.1 mirid/hill) in kharif 2020 (Table 6). The highest mirid bug population was observed after the crops+flowers treatment (1.4 ± 0.4 mirid/hill), which was at par with the population observed after the treatments with crops and flowers alone (Table 6). Overall, in both kharif seasons, the abundance of the mirid bug population was on the higher side in the rice plots surrounded with crops, flowers, and crops+flowers. In the present study, the rove beetle population of the rice fields was less abundant in both seasons. Its population exhibited no significant difference across the treatments and weeks in the kharif seasons. However, in
**Table 3 Spider population/hill in ecologically engineered rice fields during kharif 2019.**

| Treatment | Spider population[*] | | | | | | | | |
| --- | --- | --- | --- | --- | --- | --- | --- | --- | --- |
| | 44 DAT 36SMW | 54 DAT 37 SMW | 64 DAT 39 SMW | 74 DAT 40 SMW | 84 DAT 42 SMW | 94 DAT 43 SMW | 104 DAT 45 SMW | 114 DAT 46 SMW | Mean ±SE |
| T1 Crops (sesamum+ sunflower+ soybean) | 1.2 ± 0.1 | 2.1 ± 0.3 | 2.3 ± 0.2 | 2.1 ± 0.1 | 2.3 ± 0.1 | 2.6 ± 0.0 | 3.5 ± 0.3 | 3.5 ± 0.2 | 2.4 ± 0.3 |
| | (1.5 ± 0.0) | (1.8 ± 0.1) | (1.8 ± 0.1) | (1.8 ± 0.0) | (1.8 ± 0.0) | (1.9 ± 0.0) | (2.1 ± 0.1) | (2.1 ± 0.0) | (1.8 ± 0.1)[a] |
| T2 Flowers (marigold+balsam+ gaillardia) | 0.9 ± 0.1 | 1.6 ± 0.3 | 2.1 ± 0.4 | 2.1 ± 0.4 | 1.5 ± 0.3 | 2.6 ± 0.3 | 2.9 ± 0.2 | 2.7 ± 0.4 | 2.0 ± 0.2 |
| | (1.4 ± 0.0) | (1.6 ± 0.1) | (1.7 ± 0.1) | (1.7 ± 0.1) | (1.6 ± 0.1) | (1.9 ± 0.1) | (2.0 ± 0.0) | (1.9 ± 0.1) | (1.7 ± 0.1)[b] |
| T3 Natural weeds | 1.0 ± 0.3 | 1.7 ± 0.4 | 1.7 ± 0.1 | 1.7 ± 0.3 | 1.3 ± 0.1 | 1.8 ± 0.5 | 2.5 ± 0.4 | 1.8 ± 0.4 | 1.7 ± 0.2 |
| | (1.4 ± 0.1) | (1.6 ± 0.1) | (1.6 ± 0.0) | (1.6 ± 0.1) | (1.5 ± 0.0) | (1.7 ± 0.1) | (1.9 ± 0.1) | (1.6 ± 0.1) | (1.6 ± 0.0)[c] |
| T4 Crops+flowers | 1.3 ± 0.2 | 1.8 ± 0.1 | 2.1 ± 0.1 | 2.1 ± 0.1 | 2.0 ± 0.3 | 3.0 ± 0.1 | 3.9 ± 0.3 | 3.4 ± 0.2 | 2.5 ± 0.3 |
| | (1.5 ± 0.1) | (1.7 ± 0.0) | (1.8 ± 0.0) | (1.8 ± 0.0) | (1.7 ± 0.1) | (2.0 ± 0.0) | (2.2 ± 0.1) | (2.1 ± 0.1) | (1.8 ± 0.1)[a] |
| T5 Control | 1.3 ± 0.3 | 1.2 ± 0.3 | 2.1 ± 0.4 | 1.8 ± 0.2 | 1.2 ± 0.1 | 1.9 ± 0.4 | 3.3 ± 0.6 | 2.0 ± 0.1 | 1.8 ± 0.2 |
| | (1.5 ± 0.1) | (1.5 ± 0.1) | (1.8 ± 0.1) | (1.7 ± 0.1) | (1.5 ± 0.0) | (1.7 ± 0.1) | (2.0 ± 0.1) | (1.7 ± 0.0) | (1.7 ± 0.1)[b] |
| Mean ±SE | 1.1 ± 0.1 | 1.7 ± 0.1 | 2.1 ± 0.1 | 1.9 ± 0.1 | 1.7 ± 0.2 | 2.4 ± 0.2 | 3.2 ± 0.2 | 2.7 ± 0.4 | |
| | (1.5 ± 0.0)[f] | (1.6 ± 0.0)[e] | (1.7 ± 0.0)[d] | (1.7 ± 0.0)[d] | (1.6 ± 0.1)[e] | (1.8 ± 0.1)[c] | (2.0 ± 0.1)[a] | (1.9 ± 0.1)[b] | |

**Notes.**

Treatments, F = (14.7), LSD = (0.07), $P < 0.001$.
Weeks, F = (29.6), LSD = (0.09), $P < 0.001$.
Interactions, F = (1.16), LSD = (NA), $P = 0.284$.
Spider count with different superscript differ significantly.
[*]Average of ten replications.
Numbers in parenthesis are SQRT (X+1) values, SMW-Standard Meteorological Wek, DAT-Days After Transplanting.

**Table 4 Spider population/hill in ecologically engineered rice fields during kharif 2020.**

| Treatment | Spider population[*] | | | | | | | |
| --- | --- | --- | --- | --- | --- | --- | --- | --- |
| | 41 DAT 36 SMW | 51 DAT 38 SMW | 61 DAT 39 SMW | 71 DAT 41 SMW | 82 DAT 42 SMW | 91 DAT 44 SMW | 101 DAT 45 SMW | Mean ±SE |
| T1 Crops (sesamum+ sunflower+ soybean) | 2.0 ± 0.3 | 2.9 ± 0.3 | 2.5 ± 0.2 | 2.3 ± 0.3 | 3.1 ± 0.3 | 2.7 ± 0.2 | 1.4 ± 0.2 | 2.4 ± 0.2 |
| | (1.7 ± 0.1) | (2.0 ± 0.1) | (1.9 ± 0.1) | (1.8 ± 0.1) | (2.0 ± 0.1) | (1.9 ± 0.0) | (1.5 ± 0.1) | (1.8 ± 0.1)[a] |
| T2 Flowers (marigold + balsam + gaillardia) | 2.0 ± 0.3 | 2.3 ± 0.2 | 2.0 ± 0.2 | 2.4 ± 0.4 | 2.2 ± 0.1 | 2.3 ± 0.3 | 1.1 ± 0.1 | 2.0 ± 0.2 |
| | (1.7 ± 0.1) | (1.8 ± 0.1) | (1.7 ± 0.0) | (1.8 ± 0.1) | (1.8 ± 0.0) | (1.8 ± 0.1) | (1.4 ± 0.0) | (1.7 ± 0.1)[b] |
| T3 Natural weeds | 1.1 ± 0.1 | 1.2 ± 0.1 | 1.3 ± 0.1 | 1.9 ± 0.3 | 1.5 ± 0.1 | 1.6 ± 0.1 | 0.7 ± 0.2 | 1.3 ± 0.1 |
| | (1.5 ± 0.0) | (1.5 ± 0.0) | (1.5 ± 0.0) | (1.7 ± 0.1) | (1.6 ± 0.0) | (1.6 ± 0.0) | (1.3 ± 0.1) | (1.5 ± 0.0)[c] |
| T4 Crops+Flowers | 1.5 ± 0.3 | 2.1 ± 0.1 | 2.7 ± 0.3 | 2.6 ± 0.1 | 2.8 ± 0.4 | 2.2 ± 0.0 | 2.0 ± 0.3 | 2.3 ± 0.2 |
| | (1.6 ± 0.1) | (1.8 ± 0.0) | (1.9 ± 0.1) | (1.9 ± 0.0) | (1.9 ± 0.1) | (1.8 ± 0.0) | (1.7 ± 0.1) | (1.8 ± 0.1)[a] |
| T5 Control | 1.2 ± 0.1 | 0.9 ± 0.2 | 1.3 ± 0.2 | 1.4 ± 0.2 | 1.5 ± 0.2 | 1.2 ± 0.1 | 0.7 ± 0.1 | 1.1 ± 0.1 |
| | (1.5 ± 0.0) | (1.4 ± 0.1) | (1.5 ± 0.1) | (1.5 ± 0.1) | (1.6 ± 0.1) | (1.5 ± 0.0) | (1.3 ± 0.0) | (1.5 ± 0.0)[c] |
| Mean ±SE | 1.6 ± 0.2 | 1.8 ± 0.4 | 1.9 ± 0.3 | 2.1 ± 0.2 | 2.2 ± 0.3 | 2.0 ± 0.3 | 1.2 ± 0.2 | |
| | (1.6 ± 0.1)[d] | (1.7 ± 0.1)[b] | (1.7 ± 0.1)[b] | (1.8 ± 0.1)[a] | (1.8 ± 0.1)[a] | (1.7 ± 0.1)[b] | (1.5 ± 0.1)[c] | |

**Notes.**

Treatments, F =(47.9), LSD =(0.06), $P < 0.001$.
Weeks, F =(13.8), LSD =(0.08), $P < 0.001$.
Interactions, F = (1.6), LSD = (0.18), $P = 0.044$.
Spider count with different superscript differ significantly.
[*]Average of ten replications.
Numbers in parenthesis are SQRT (X+1) values, SMW-Standard Meteorological Wek, DAT-Days After Transplanting.

Yele et al. (2023), *PeerJ*, DOI 10.7717/peerj.15531

**Table 5  Mirid bug population/hill in ecologically engineered rice fields during kharif 2019.**

| Treatments | Mirid bug population[*] | | | | | | | | |
|---|---|---|---|---|---|---|---|---|---|
| | 44 DAT 36SMW | 54 DAT 37 SMW | 64 DAT 39 SMW | 74 DAT 40 SMW | 84 DAT 42 SMW | 94 DAT 43 SMW | 104 DAT 45 SMW | 114 DAT 46 SMW | Mean ±SE |
| **T1** Crops (sesamum+ sunflower+soybean) | 0.3 ± 0.1 | 1.3 ± 0.2 | 0.2 ± 0.2 | 0.1 ± 0.0 | 0.1 ± 0.1 | 0.2 ± 0.2 | 12.5 ± 1.0 | 11.5 ± 0.7 | 3.3 ± 1.9 |
| | (1.1 ± 0.1) | (1.5 ± 0.1) | (1.1 ± 0.1) | (1.0 ± 0.0) | (1.0 ± 0.0) | (1.1 ± 0.1) | (3.7 ± 0.1) | (3.5 ± 0.1) | (1.8 ± 0.4)[a] |
| **T2** Flowers (marigold+balsam+gaillardia) | 0.3 ± 0.1 | 0.7 ± 0.2 | 0.4 ± 0.1 | 0.0 ± 0.0 | 0.1 ± 0.0 | 0.4 ± 0.2 | 12.1 ± 1.3 | 12.4 ± 0.6 | 3.3 ± 2.0 |
| | (1.1 ± 0.1) | (1.3 ± 0.1) | (1.2 ± 0.0) | (1.0 ± 0.0) | (1.0 ± 0.0) | (1.2 ± 0.1) | (3.6 ± 0.2) | (3.7 ± 0.1) | (1.8 ± 0.4)[a] |
| **T3** Natural weeds | 0.3 ± 0.1 | 0.8 ± 0.1 | 0.1 ± 0.0 | 0.0 ± 0.0 | 0.1 ± 0.1 | 0.3 ± 0.2 | 5.8 ± 1.5 | 4.4 ± 1.3 | 1.5 ± 0.8 |
| | (1.1 ± 0.1) | (1.3 ± 0.0) | (1.0 ± 0.0) | (1.0 ± 0.0) | (1.1 ± 0.0) | (1.1 ± 0.1) | (2.6 ± 0.3) | (2.3 ± 0.3) | (1.4 ± 0.2)[b] |
| **T4** Crops+Flowers | 0.6 ± 0.4 | 0.6 ± 0.3 | 0.3 ± 0.1 | 0.1 ± 0.1 | 0.1 ± 0.1 | 0.4 ± 0.1 | 10.9 ± 0.6 | 10.0 ± 0.7 | 2.8 ± 1.7 |
| | (1.2 ± 0.1) | (1.3 ± 0.1) | (1.1 ± 0.1) | (1.0 ± 0.0) | (1.0 ± 0.0) | (1.2 ± 0.0) | (3.4 ± 0.1) | (3.3 ± 0.1) | (1.7 ± 0.4)[a] |
| **T5** Control | 0.3 ± 0.2 | 0.8 ± 0.3 | 0.1 ± 0.1 | 0.1 ± 0.1 | 0.3 ± 0.2 | 0.5 ± 0.4 | 8.9 ± 1.1 | 3.4 ± 1.0 | 1.8 ± 1.1 |
| | (1.1 ± 0.1) | (1.3 ± 0.1) | (1.0 ± 0.0) | (1.0 ± 0.0) | (1.1 ± 0.1) | (1.2 ± 0.1) | (3.1 ± 0.2) | (2.1 ± 0.2) | (1.5 ± 0.3)[b] |
| Mean ±SE | 0.3 ± 0.1 | 0.8 ± 0.1 | 0.2 ± 0.1 | 0.0 ± 0.0 | 0.1 ± 0.0 | 0.3 ± 0.0 | 10.0 ± 1.2 | 8.3 ± 1.9 | |
| | (1.1 ± 0.0)[c] | (1.3 ± 0.0)[b] | (1.1 ± 0.0)[c] | (1.0 ± 0.0)[c] | (1.1 ± 0.0)[c] | (1.1 ± 0.0)[c] | (3.3 ± 0.2)[a] | (3.0 ± 0.3)[a] | |

**Notes.**

Treatments, F =(16.2), LSD =(0.10), $P < 0.001$.

Weeks, F =(373.6), LSD = (0.13), $P < 0.001$.

Interactions, F = (7.13), LSD = (0.3), $P < 0.001$.

Mirid bug count with different superscript differ significantly.

[*]Average of ten replications.

Numbers in parenthesis are SQRT (X+1) values, SMW-Standard Meteorological Wek, DAT-Days After Transplanting.

**Table 6  Mirid bug population/hill in ecologically engineered rice fields during kharif 2020.**

| Treatments | Mirid Bug population* | | | |
| | 82 DAT 42 SMW | 91 DAT 44 SMW | 101 DAT 45 SMW | Mean ±SE |
|---|---|---|---|---|
| **T1** Crops (sesamum+ sunflower+ soybean) | 1.5 ± 0.5 (1.6 ± 0.2) | 0.7 ± 0.1 (1.3 ± 0.1) | 0.9 ± 0.1 (1.4 ± 0.0) | 1.0 ± 0.3 (1.4 ± 0.1)[a] |
| **T2** Flowers (marigold+balsam+ gaillardia) | 1.5 ± 0.3 (1.6 ± 0.1) | 1.0 ± 0.3 (1.4 ± 0.1) | 0.9 ± 0.2 (1.4 ± 0.1) | 1.1 ± 0.2 (1.4 ± 0.1)[a] |
| **T3** Natural weeds | 0.9 ± 0.5 (1.3 ± 0.2) | 0.2 ± 0.2 (1.1 ± 0.1) | 0.5 ± 0.1 (1.2 ± 0.0) | 0.6 ± 0.2 (1.2 ± 0.1)[b] |
| **T4** Crops+flowers | 2.2 ± 0.5 (1.8 ± 0.1) | 1.0 ± 0.1 (1.4 ± 0.0) | 1.2 ± 0.2 (1.5 ± 0.1) | 1.4 ± 0.4 (1.5 ± 0.1)[a] |
| **T5** Control | 0.5 ± 0.2 (1.2 ± 0.1) | 0.3 ± 0.2 (1.1 ± 0.1) | 0.6 ± 0.1 (1.3 ± 0.0) | 0.5 ± 0.1 (1.2 ± 0.0)[b] |
| Mean ±SE | 1.3 ± 0.2 (1.5 ± 0.1)[a] | 0.6 ± 0.1 (1.3 ± 0.1)[b] | 0.8 ± 0.1 (1.3 ± 0.0)[b] | |

**Notes.**

Treatments, F =(7.24), LSD =(0.15), $P < 0.001$.
Weeks, F =(7.33), LSD = (0.12), $P < 0.001$.
Interactions, F = (0.5), LSD = (NA), $P = 0.780$.
Mirid bug count with different superscript differ significantly.
*Average of ten replications.
Numbers in parenthesis are SQRT (X+1) values, SMW-Standard Meteorological Wek, DAT-Days After Transplanting.

kharif 2019, it appeared early in the season, that is, at 44 DAT, whereas in kharif 2020, it appeared at 61 DAT (Tables 7 & 8). In both the kharif seasons, natural enemy populations were more abundant in the rice plots planted with crops, flowers, and crops+flowers than in the control plots. Among the natural enemies in rice crops, the spider population was more abundant than the mirid bug and rove beetle populations.

## Attraction response of spiders in the Y-tube olfactometer

The olfactory response of spiders toward leaves of plant species, namely sesamum, balsam, sunflower, marigold, and soybean, was studied using a Y-tube olfactometer. Sesamum ($P < 0.01$) and balsam ($P < 0.05$) attracted considerably more spiders than the other plant species. The spiders exhibited the highest attraction toward sesamum leaves (83.3% ± 1.67%), followed by balsam (73.33% ± 3.33%) (Fig. 3). Spiders also exhibited attraction toward sunflower and marigold leaves but the attraction was not statistically significant. The present study revealed that spiders were attracted more toward sesamum and balsam leaves than toward the other plants.

## Rice grain yield in ecologically engineered rice fields

The rice yield differed significantly between the treatments in 2020 ($F = 25.2$, $P < 0.001$), but the difference was less significant during 2019 ($F = 3.7$, $P = 0.033$) (Table 8). However, rice yield was substantially higher in rice plots planted with oilseed crops+flowers during 2019 (5.60 ± 0.24 tons/ha) and 2020 (5.27 ± 0.06 tons/ha) than in the control rice plots. Treatments with oilseed crops alone (4.52 ± 0.24 and 4.71 ± 0.07 tons/ha) and flowers

**Table 7  Rove beetle population/hill in ecologically engineered rice fields during kharif 2019.**

| Treatments | Rove beetle population[*] | | | | | | Mean ±SE |
|---|---|---|---|---|---|---|---|
| | 44 DAT 36SMW | 54 DAT 37 SMW | 64 DAT 39 SMW | 74 DAT 40 SMW | 84 DAT 42 SMW | 94 DAT 43 SMW | |
| **T1** Crops (sesamum+ sunflower+ soybean) | 0.2 ± 0.1 | 0.3 ± 0.2 | 0.1 ± 0.0 | 0.0 ± 0.0 | 0.3 ± 0.1 | 0.6 ± 0.2 | 0.2 ± 0.1 |
| | (1.1 ± 0.1) | (1.1 ± 0.1) | (1.0 ± 0.0) | (1.0 ± 0.0) | (1.1 ± 0.0) | (1.3 ± 0.1) | (1.1 ± 0.0)[a] |
| **T2** Flowers (marigold+balsam+ gaillardia) | 0.0 ± 0.0 | 0.2 ± 0.1 | 0.1 ± 0.0 | 0.2 ± 0.1 | 0.3 ± 0.1 | 0.3 ± 0.1 | 0.2 ± 0.0 |
| | (1.0 ± 0.0) | (1.1 ± 0.0) | (1.0 ± 0.0) | (1.1 ± 0.0) | (1.1 ± 0.0) | (1.1 ± 0.0) | (1.1 ± 0.0)[a] |
| **T3** Natural weeds | 0.0 ± 0.0 | 0.1 ± 0.0 | 0.1 ± 0.1 | 0.1 ± 0.0 | 0.1 ± 0.0 | 0.3 ± 0.1 | 0.1 ± 0.0 |
| | (1.0 ± 0.0) | (1.0 ± 0.0) | (1.0 ± 0.0) | (1.0 ± 0.0) | (1.0 ± 0.0) | (1.1 ± 0.0) | (1.0 ± 0.0)[a] |
| **T4** Crops+flowers | 0.1 ± 0.1 | 0.1 ± 0.0 | 0.3 ± 0.1 | 0.1 ± 0.0 | 0.2 ± 0.1 | 0.3 ± 0.2 | 0.2 ± 0.0 |
| | (1.1 ± 0.0) | (1.1 ± 0.0) | (1.1 ± 0.0) | (1.0 ± 0.0) | (1.1 ± 0.1) | (1.1 ± 0.1) | (1.1 ± 0.0)[a] |
| **T5** Control | 0.1 ± 0.1 | 0.1 ± 0.0 | 0.1 ± 0.0 | 0.1 ± 0.0 | 0.1 ± 0.0 | 0.2 ± 0.1 | 0.1 ± 0.0 |
| | (1.0 ± 0.0) | (1.0 ± 0.0) | (1.1 ± 0.0) | (1.1 ± 0.0) | (1.0 ± 0.0) | (1.1 ± 0.1) | (1.1 ± 0.0)[a] |
| Mean ±SE | 0.1 ± 0.0 | 0.1 ± 0.0 | 0.1 ± 0.0 | 0.1 ± 0.0 | 0.2 ± 0.0 | 0.3 ± 0.1 | |
| | (1.0 ± 0.0)[c] | (1.1 ± 0.0)[b] | (1.1 ± 0.0)[b] | (1.1 ± 0.0)[b] | (1.1 ± 0.0)[b] | (1.2 ± 0.0)[a] | |

**Notes.**

Treatments, F =(2.2), LSD =(NA), $P = 0.075$.
Weeks, F =(5.19), LSD = (0.05), $P = 0.003$.
Interactions, F = (1.02), LSD = (NA), $P = 0.438$.
Rove beetle count with different superscript differ significantly.
[*]Average of ten replications.
Numbers in parenthesis are SQRT (X+1) values, SMW-Standard Meteorological Wek, DAT-Days After Transplanting.

**Table 8  Rove beetle population/hill in ecologically engineered rice fields during kharif 2020.**

| Treatments | Rove beetle population[*] | | | | Mean ±SE |
|---|---|---|---|---|---|
| | 61 DAT 39 SMW | 71 DAT 41 SMW | 82 DAT 42 SMW | 91 DAT 44 SMW | |
| **T1** Crops (sesamum+ sunflower+ soybean) | 0.4 ± 0.1 | 0.2 ± 0.1 | 0.3 ± 0.1 | 0.1 ± 0.0 | 0.2 ± 0.1 |
| | (1.2 ± 0.0) | (1.1 ± 0.0) | (1.1 ± 0.0) | (1.0 ± 0.0) | (1.1 ± 0.0)[a] |
| **T2** Flowers (marigold+ balsam+ gaillardia) | 0.3 ± 0.1 | 0.3 ± 0.2 | 0.1 ± 0.0 | 0.0 ± 0.0 | 0.2 ± 0.1 |
| | (1.1 ± 0.0) | (1.1 ± 0.1) | (1.0 ± 0.0) | (1.0 ± 0.0) | (1.1 ± 0.0)[a] |
| **T3** Natural weeds | 0.2 ± 0.1 | 0.1 ± 0.0 | 0.2 ± 0.1 | 0.0 ± 0.0 | 0.1 ± 0.0 |
| | (1.1 ± 0.0) | (1.0 ± 0.0) | (1.1 ± 0.0) | (1.0 ± 0.0) | (1.1 ± 0.0)[a] |
| **T4** Crops+flowers | 0.3 ± 0.1 | 0.2 ± 0.0 | 0.2 ± 0.1 | 0.0 ± 0.0 | 0.2 ± 0.1 |
| | (1.1 ± 0.1) | (1.1 ± 0.0) | (1.1 ± 0.0) | (1.0 ± 0.0) | (1.1 ± 0.0)[a] |
| **T5** Control | 0.1 ± 0.1 | 0.3 ± 0.1 | 0.1 ± 0.0 | 0.0 ± 0.0 | 0.1 ± 0.0 |
| | (1.0 ± 0.0) | (1.1 ± 0.1) | (1.0 ± 0.0) | (1.0 ± 0.0) | (1.1 ± 0.0)[a] |
| Mean ±SE | 0.3 ± 0.0 | 0.2 ± 0.0 | 0.2 ± 0.0 | 0.1 ± 0.0 | |
| | (1.1 ± 0.0)[a] | (1.1 ± 0.0)[a] | (1.1 ± 0.0)[a] | (1.0 ± 0.0)[b] | |

**Notes.**

Treatments, F =(1.39), LSD =(NA), $P = 0.247$.
Weeks, F =(6.38), LSD = (0.04), $P < 0.001$.
Interactions, F = (0.815), LSD = (NA), $P = 0.633$.
Rove beetle count with different superscript differ significantly.
[*]Average of ten replications.
Numbers in parenthesis are SQRT (X+1) values, SMW-Standard Meteorological Wek, DAT-Days After Transplanting.

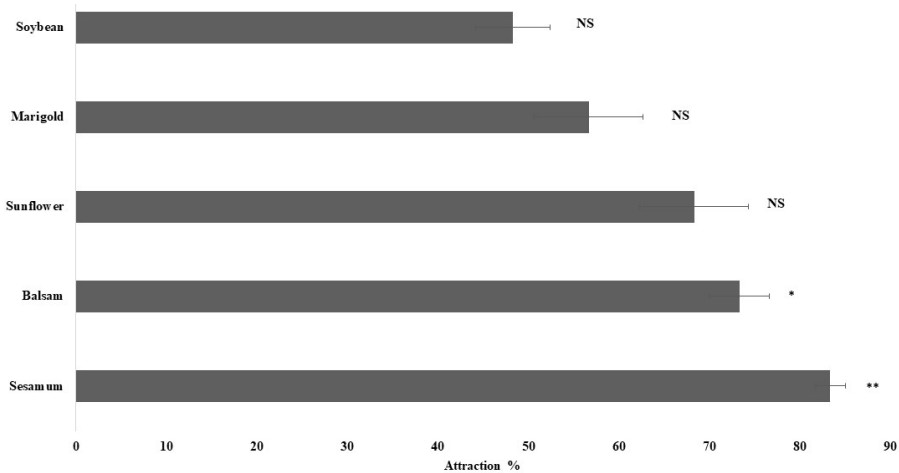

**Figure 3 Orientation responses of spiders towards flowering plants through Y-tube olfactometer.** Error bars represents standard error of mean percent spider response. Error bars represents standard error of mean percent spider response. Bars denoted by asterisks indicate a significant response for the flowering plants (* $p < 0.05$, ** $P < 0.01$, NS- non-significant; two-tailed paired $t$-test).

alone ($4.80 \pm 0.55$ and $4.97 \pm 0.05$ tons/ha) recorded significantly higher yields than the control treatment in both seasons (Fig. 4).

## DISCUSSION

Over the past few decades, integrated pest management (IPM) has become a way of life and has shown a great potential for reducing the dependence on chemical-based control methods (*Pretty, 1998*; *Atanassov et al., 2002*). Ecological engineering should be treated as a refined and precise version of IPM in agricultural ecosystems. IPM requires coordinated efforts integrating diverse tactics, including cultural, biological, and chemical control (*Dent, 1991*). Ecological engineering strategies for pest management involve the use of vegetation management-based cultural practices to enhance biological control or the bottom-up effect that acts directly on pests (*Horgan et al., 2016*). These strategies involve identifying optimal forms of botanical diversity and incorporating them into a farming system so as to suppress pests by promoting their natural enemies. We attempted to use diverse plant species, including flowering annuals, around the rice crop to study their effect on the occurrence of different rice pests and the abundance of natural enemies for two successive kharif seasons.

In both seasons, the rice plots planted with oilseed crop plants, flowering plants, and combinations of crops and flowering plants exhibited lower abundance of the BPH population than the general rice fields. Moreover, the peak BPH population appeared higher and earlier in the season in the conventional rice plots than in the ecologically engineered rice plots. Furthermore, the pest population build-up was slower in the rice plots planted with oilseed crops and flowering annuals than in the conventional rice plots. The year 2019 recorded the lowest BPH population in the rice plots planted with crops

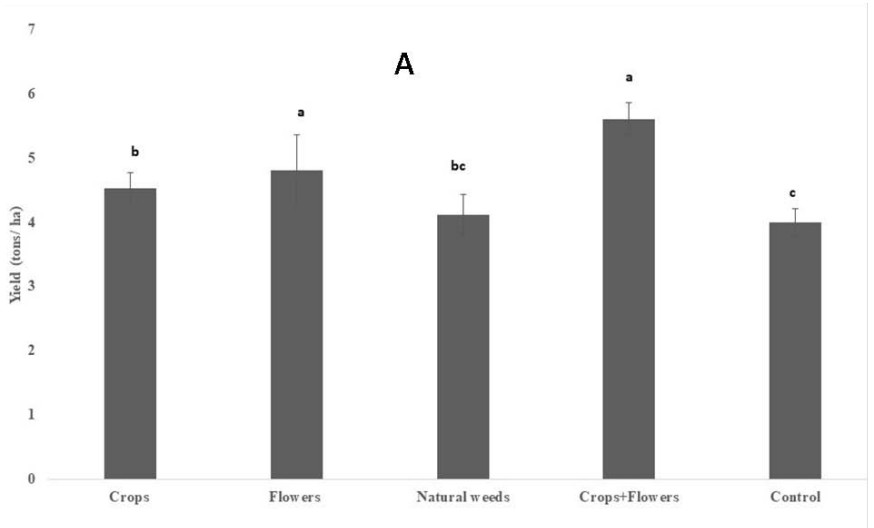

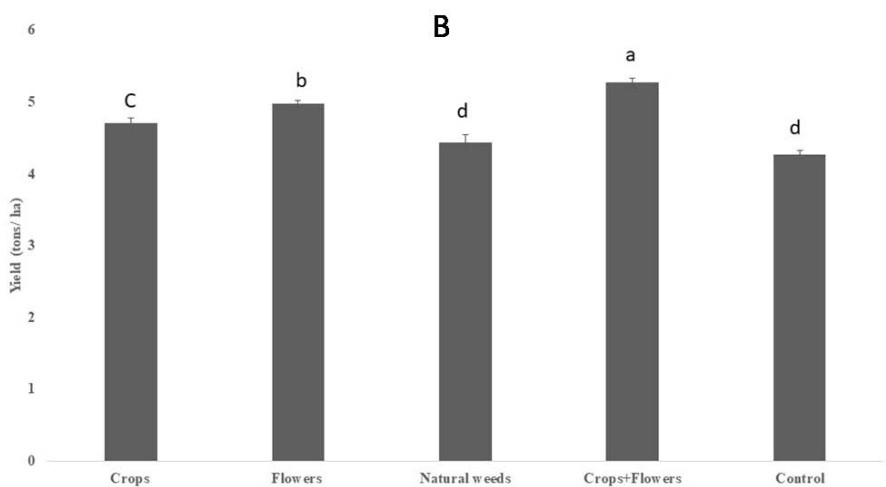

**Figure 4** Rice grain yield in ecologically engineered rice fields during kharif 2019 (A) and 2020 (B).

such as sesamum, sunflower, and soybean. However, in 2020, the lowest BPH population was observed in the rice plots planted with oilseed crops and flowering plants. This suggests that rice crops grown with diverse crops and flowering plants had a lower BPH population than the conventionally grown rice crops.

Natural enemies such as spiders, mirid bugs, and rove beetles are not strictly specialized predators that prey on leafhoppers, planthoppers, and soft-bodied caterpillar pests. The abundance of these natural enemies helps naturally suppress many crop pests in rice fields, especially Hemipteran pests such as planthoppers and leafhoppers. The spider population was abundant in all the ecologically engineered plots throughout both kharif seasons. Rice plots planted with oilseed crops and flowers doubled the antagonist's population in

the control plots during the kharif seasons. Notably, such rice fields had greater spider abundance than those subjected to other treatments and the control plots.

We found greater mirid bug abundance during kharif 2019, which was also reported early in the season. On the other hand, during kharif 2020, the abundance of mirid bugs was lower and reported later in the season, which may be due to the late BPH appearance. During both seasons, the mirid bug population was considerably more abundant in the ecologically engineered plots than in the plots treated with natural weeds and control plots. The mean mirid population was twice as abundant as in the ecologically engineered plots or control rice plots. Similarly, the rove beetle population was higher in the rice plots planted with oilseed crops and flowering plants. The population of natural enemies (spiders, mirid bugs and rove beetles) was more abundant in the rice plots planted with crops, flowers, and crops+flowers. The spider population was more abundant in the rice fields than the mirid bug and rove beetle populations.

Furthermore, olfactory response studies with the Y-tube suggested that spiders are more attracted to sesamum and balsam plant leaves. Sesamum and balsam leaves were, therefore, better spider attractants. The rice yield was higher in the plots subjected to the ecologically engineered treatments than the weedy and control plots. During both seasons, the highest yield originated from plots planted with the crop and flowering plant combinations. Thus, rice fields planted with flowering and other crop plants had lower pest activity; consequently, this strategy reduced the damage caused by insect pests and enhanced the rice crop yield.

Reduced pest activity and delayed BPH appearance in the rice growing season in ecologically engineered rice fields may be related to the higher activities of natural enemies in the diverse crop and flowering plant system around the rice crop. An array of vegetation present around the main crop provides shelter and floral resources in the form of nectar food for the natural enemies. Staggered planting of flowering plants ensures that the flowers are available for a longer duration in the rice-growing season. It also broadens the availability of floral resources and food for the natural enemies. Increased flowering plant diversity around the rice crop field may positively increase the activity of natural enemies and help suppress the pest population. Higher activities of natural enemies such as spiders and mirid bugs in the ecologically engineered plots may control the BPH population and slow down population build-up throughout the rice season. Similar results have been reported in some initial ecological engineering studies (*Yu, Barrion & Lu, 2001*; *Gurr et al., 2011*; *Liu et al., 2014*). Flowering plants inundated around the main crops reduce the pest population by enhancing the activities of natural enemies (*Zhu et al., 2015*; *Chen et al., 2016*; *Kong et al., 2016*; *Keerthi et al., 2016*). The presence of grasses and weed flora around rice fields (*Chen et al., 2016*) and the planting of sesamum crops on bunds as a nectar source (*Zhu et al., 2015*; *Yele et al., 2022*) together reduce pest abundance in the rice fields.

Similarly, intercropping zizania, planting vetiver grass along irrigation canals, and releasing *Trichogramma* lowered the pest activity in rice fields (*Zhu et al., 2017*). Ecological engineering techniques have increased the activity of egg parasitoids of planthoppers such as *Oligosita* and *Anagrus*, thereby significantly reducing in the planthoppers' population in rice (*Zhu et al., 2015*).

Planting sesame around the rice plots is a known measure for improving the activities of natural enemies in these plots, which helps suppress pest populations. Planting flowering plants, such as marigolds, balsam, and gaillardia, around the rice plots helps attract natural enemies by providing them with nectar and a harboring/resting place around the rice fields. *Zhu et al. (2014)* proposed that flowering plants around the rice, such as *T. erecta*, *T. procumbens*, *E. sonchifolia*, and *S. indicum*, reduce the planthopper pest population and increase the abundance of natural enemies such as mirid bugs. Predation efficiency and consumption of BPH by *C. lividipennis* increased in the flower treatment plots. *S. indicum* was the most favorable flowering plant and strongly promoted *C. lividipennis* predation. These results align with those of the present study and suggested that *S. indicum* is well suited for ecological engineering on bunds of rice crops. *Anagrus* spp. and *A. nilaparvatae* are egg parasitoids that have a relevant role in managing leaf-hoppers and planthoppers (*Yu, Barrion & Lu, 2001*; *Gurr et al., 2011*).

Our findings complement the findings of previous studies exhibiting that ecological engineering technology can maintain pest populations in rice at lower levels than conventional cultivation throughout the rice-growing season without hampering the yield.

Olfactometer studies have revealed that volatile compounds emitted by plant species such as *S. indicum*, *I. balsamina*, *E. sonchifolia*, *T. procumbens*, and *H. esculentus* attract *Anagrus* spp. Parasitoids also enhance their biological performance (*Zhu et al., 2013*). Notably, ample access to sesame flowers enhances the lifespan of *A. nilaparvatae* and *A. optabilis*. Similarly, our olfactometer study also revealed that spiders exhibited highest attraction toward sesamum and balsam leaves. The volatile compounds released by these plants may attract spiders toward the plants. Ample availability of nectar food for bioagents can improve the reproductive abilities of natural enemies as well as the survival and host-searching ability. Nectar food plays a pivotal role in enhancing the biological control ability of natural enemies (*Wackers, Van Rijn & Bruin, 2005*; *Poddar, Yele & Kumari, 2019*). Planting nectar and floral resources such as sesame, marigold, and sunflower in the crop's near vicinity is effectively improves the biological control and conservation of biocontrol agents (*Lu & Guo, 2015*).

Access to sesamum flowers significantly enhances the longevity of adult parasitoid and the parasitization rate on BPH eggs (*Zhu et al., 2012*). Likewise, this access significantly increases the fecundity of the egg parasitoid *Trichogramma chilonis* on lepidopterous pests. Sesamum flowers also improve the longevity of egg parasitoids of lepidopterous pests such as pink stem borers, spotted stem borers and leaf folders without hosting these pests (*Zhu et al., 2012*; *Zhu et al., 2015*).

## CONCLUSION

Vegetation around the field significantly enhances the structural and functional diversity of arthropods. The availability of structural habitats as vegetative growth of crops, floral resources, longer flowering duration, and nectar provided by diverse vegetation greatly improves the activity of natural enemies. Higher activities of natural enemies in the

ecologically engineered fields directly affect the incidence and population build-up of *N. lugens*. Ecological engineering aims to reduce pest-induced damage by maximizing natural mortality through the strategic introduction of plant diversity. Ecological engineering has a great potential and is developing rapidly as a fully available and effective biological pest control method within the IPM strategy. Planting oilseed crop plants such as sesamum, sunflower, and soybean and flowering crops such as marigold, balsam, and gaillardia on the bunds around the main rice field alone or in combination enhances the activity of natural enemies, thereby allowing the management of the rice infesting *N. lugens* population. As the awareness about the effects of insecticides is growing among farmers, ecological engineering paves the way for sustainable pest management in rice. Ecological engineering techniques can reduce the pesticide use for plant protection. In today's changing climate scenario, these techniques are an ecologically sustainable option for biotic stress mitigations in climate resilient agriculture. Moreover, ecological engineering needs to be promoted as a pest smart strategy for climate smart agriculture. We here recommend adopting and developing this ecological engineering technique in the IPM module for sustainable *N. lugens* management in rice. Further large-scale studies and farmer field trials are warranted for evaluating diverse flowering plants to entrench into this ecological engineering approach.

## ACKNOWLEDGEMENTS

The authors thank the Director of the ICAR-Indian Agriculture Research Institute, Pusa Campus, New Delhi for providing necessary facilities for the research.

### Funding

The authors received no funding for this work.

### Competing Interests

The authors declare there are no competing interests.

### Author Contributions

- Yogesh Yele conceived and designed the experiments, performed the experiments, analyzed the data, prepared figures and/or tables, authored or reviewed drafts of the article, and approved the final draft.
- Subhash Chander conceived and designed the experiments, performed the experiments, analyzed the data, prepared figures and/or tables, authored or reviewed drafts of the article, and approved the final draft.
- Sachin S. Suroshe conceived and designed the experiments, performed the experiments, analyzed the data, prepared figures and/or tables, authored or reviewed drafts of the article, and approved the final draft.
- Suresh Nebapure analyzed the data, prepared figures and/or tables, and approved the final draft.

- Prabhulinga Tenguri analyzed the data, prepared figures and/or tables, and approved the final draft.
- Arya Pattathanam Sundaran analyzed the data, prepared figures and/or tables, and approved the final draft.

## Data Availability

The raw data is available in the Supplemental File.

## Supplemental Information

Supplemental information for this article can be found online at http://dx.doi.org/10.7717/peerj.15531#supplemental-information.

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
