# Peer review of "Ecological engineering in low land rice for brown plant hopper, Nilaparvata lugens (Stål) management"

_PeerJ, doi:10.7717/peerj.15531_

## Round 0.1 · original submission · Minor Revisions

Dear authors,

After examining several reviews, we consider your work suitable for publication after making minor changes. It took me a long time to formulate my decision because some of the opinions were brief, not pointing out specific changes to the draft.

Please consider the referees' suggestions, particularly those related to statistical design and data representation.

The submission is good, and the study deserves publication once you consider the suggestions.

Thank you very much
Francesco

·

Basic reporting

Dear Editor
Thank you very much for the opportunity to review the manuscript on Ecological engineering in low land rice for brown plant hopper, Nilaparvata lugens management. I compliment all the authors for their wonderful basic work on ecological engineering, based on these results IPM management strategy can be developed for effective management of brown plant hopper.
There are some of my comments on the manuscript:

- In abstract remove standard error mean values
- Any synchrony in the flowering of these bund crops ,sesamum+sunflower+ soybean
- Line 127: what was the bund size or grown in pots and kept around the paddy plot.
- In tables kindly mention BPH population per hill???
- Club fig 1 and 3, club figure 2 and 4
- Kindly club the tables 6 , 7 and 8 (all predators in one table)
- In all the tables kindly remove the SQRT (X+1) values
- Reduce the results part which also depicted in the form of graphs and tables.

Experimental design

Its appropriate care has taken while designing the experiment

Validity of the findings

Its good study and found the some of the plants can be used as a bund crop to enhance the natural enemies population

Reviewer 2 ·

Basic reporting

The article has professional English, sufficient background, introduction, material and methods, results and discussions and references

Experimental design

No comments

Validity of the findings

No comments

Additional comments

No information on Yellow stem borer.
It would be better, it have given pooled data of both years also.
Still it need to know the scientific name of natural enemies

Annotated reviews are not available for download in order to protect the identity of reviewers who chose to remain anonymous.

·

Basic reporting

Authors are required to proofread the manuscript to correct some typos and grammar mistakes in the current version of the manuscript.

Experimental design

No comments.

Validity of the findings

1. The findings are interesting and data are analyzed in a proper statistical manner. I have only one suggestion. Along with the results (table and graphs) the authors should include some pictures (fields, BPH per hill situation, intercropping pictures), which will strengthen the readability of the paper and the readers can easily relate the findings.

2. Most of the data are in tables. The authors are advised to try to put more data in a graphical manner.

3. Discussion has to be improved. Please discuss more on the olfactory response results. Relate the olfactory results to the field data and derive significant conclusions.

Additional comments

Please improve the footnotes of the tables. Expand all abbreviations used in the table in the footnotes.

·

Basic reporting

Well written MS with clear cut and meaningful text, adequate literature have been reviewed in support of the research findings, adequate no of tables and figures with desired structure and quality have been provided by the authors

Experimental design

Well defined research problem with clear cut objectives and sound methods used for data analyses and interpretation

The finding are relevant to the advancement of the state of knowledge on the subject and the topic of research is well suited to the journals aims and scope

Validity of the findings

All the requisite data which is statistically sound, clear cut and meaningful has been provided by the authors

Valid conclusions drawn based on salient findings of the research and discussed in pros and cons with literature available on the topic

Additional comments

1. Page 6, line, 46
pl check if it is "underline"

2.Page 9, Line 126
Please describe what is the control you kept? I guess it is a rice crop with all agronomic practices like weeding, etc and no any banker crop/s grown around

3. Page 9, line 127
Instead of using a term "Field crops", why not to use "oilseed crops" as all the three crops are oilseeds.
Moreover it will be good to say "oilseed crops" and "flowering crops" were used for ecological engineering of rice.
If seems appropriate, pl make the changes uniformly throughout the text of MS

4. Page 10, Line No. 173-174
You did not mention, which method was / followed for carrying out Olfactometer studies using Y tube?
Kindly reefer for dtails
Fand BB*, Amala U, Yadav DS, Rathi R, Mhaske SH, Upadhyay A, Shabeer ATP, Kumbhar DS. 2020. Bacterial volatiles from mealybug honeydew exhibit kairomonal activity towards solitary endoparasitoid Anagyrus dactylopii. Journal of Pest Science. 93 (1), 195-206

5. Page No. 1o, Line No. 177-178
This part is relevant to "Materials and methods" section. Please see if its mentioned there, then you can ommit from here. Else, it can be shifted to MM section.
Please restrict your descriptions relevant to important results/ findings only in 'Results" section.

6. Page 13, line 258
Do you know what was the triggering factor for attraction of spiders towards the plant species you have tested?
Any chemical/biochemical cue mediating these interactions were figured out?


7. Page 17, line 375
What do you think that ecological Engineering important from the point of adaptation and mitigation as a component of climate resilient agriculture

Is there any significant reduction in need for chemical pesticide use for tackling pest problems in ecologically enginnered crop fields?

pl correlate your salient findings with EE technique as pest smart strategy for climate smart agriculture

Reviewer 5 ·

Basic reporting

A hypothesis needs to be drawn based on the previous literature.

All the reference except Pretty, 1998 is missing in the text.

Add the suggested Keerthi et al. 2020 in the text and reference section.

The similar result of figures 1, 2, 3, and 4 are already presented in the table, mention either in a table or in the figure.

In the discussion, each sentence must be supported by earlier findings, or contradictory results may be mentioned.

Experimental design

The introduction part in lengthy, which might be reduced

Validity of the findings

The present highlight the importance of EE in rice plant protection. The concept of EE reduces the pest populations and leaves no or low pesticide residues in the crop ecosystem.

The concept needs to be promoted and encouraged among the farming fraternity.

Additional comments

The grammatical corrections were suggested in the manuscript, which might be attended

Annotated reviews are not available for download in order to protect the identity of reviewers who chose to remain anonymous.

·

Basic reporting

Even though the Ecological engineering is not my field work, is seems that the manuscript try to fill an interesting gap regarding the effectiveness of this methodology to control Brown Planthopper (BPH), Nilaparvata lugens, pest in rice crops. The manuscript is written in a clear and unambiguous English, and seems that sufficient background is provided using appropriated references.

Although the manuscript conforms to an acceptable format of ‘standard sections’, I miss a more detailed info about the impact that Nilaparvata lugens has on rice crops to enance the importance of the present work in introduction or even discussion. I also miss an introductory part and a direct mention as an objective relating to the olfactory response studies included in this work. It was really strange found this olfactory response studies in material and methods section and in results without having any previous info regarding them. In contrast, results of the olfactory response studies described in lines 259 and 266 are redundant. The text of the paragraph should be rewritten to avoid repeated info. Additionally, along the discussion there is no mention again about the olfactory response results. This olfactory response results should be interpreted and discussed doing comparison with similar studies, giving a final recommendation following the results obtained in this part of the study. In contrast, lines 280 to 314 of the discussion section seems more a summary of results that a discussion itself. A short summary of the most important results at the beginning of the discussions section has sense, but in this case is too long. Among these lines, a more deep interpretation or discussion of each detailed result needs to be done, or alternatively I suggest eliminate this part.

Tables and figures should be improved. Why in tables 1-8 are showed data with and without square root transformation? It is not explained anywhere. All table captions should include the complete definition of DAT, SMW and the superscript letters included in some values. In these tables, least significant differences (LSD) results included has no sense, because it should appear at least one value (statistic and p-value) per each pair of treatments compared in the a posteriori test and not just one. However, the data showed in these tables could be better explained using figures instead, and just keep tables to show the statistical results and not the means of treatments and weeks. In this sense, figures 1-4 should include the mean of each week for all treatments to identify easier the trends of the data. Additionally, the superscripts letters of the a posterior test could also be included in the figures to better understand the significant differences among groups. To be consistent with the order used in the text to explaining the results, figures 5 and 6 should be interchanged. In the raw data shared, there is no info regarding the olfactory response studies; this data should be included in the file in accordance with the Data Sharing policy of the journal.

Experimental design

The manuscript clearly define the research question, and fits within the aims and scope of the journal. However, as I does not work on ecological engineering I cannot assess whether this research question is relevant, meaningful, and fills an identified knowledge gap of the field.

However, considering my data analysis background, in my opinion the investigation was not conducted rigorously and to a high technical standard. In some parts methods were not described with sufficient information to be reproducible by another investigator, and statistical analysis were not applied correctly. The experimental design and methodology applied during the experiments in the field were correctly explained. But there are some inconsistencies with the data reported in the raw data file and the statistical analysis applied. I will describe in detail some of the issues found.

Regarding the study on BPH and their natural enemy population, if I have not misunderstood, the experimental design incorporates several factors: treatment (5 different treatments combining crops, flowes, weeds and control), plots (4 replicates per each treatment), weeks (different times (7-8) in were variables were observed) and year (the experiment was conducted twice in 2019 and 2020). In each plot, 10 hills were selected and tagged randomly; these tagged hills were used for further data recording. One of the main issues that I found is that for statistical analysis, only were considered two factors (treatment and week) in a two-way ANOVA (vaguely explained in lines 144-149), missing the other sources of variability. Plots and year should be consider in the statistical analysis as random factors to avoid temporal and spatial pseudoreplication.

The authors mentioned in results sections some sentences like “Overall, BPH population was higher during kharif 2020 as compared to kharif 2019, irrespective of the treatments “ (line 201), “In general, there was no difference in the abundance of spider populations during kharif 2019 and 2020.” (line 210) or “The mirid bug population abundance was higher in kharif 2019 as compared to kharif 2020.” (line 228), but they never compare statistically data between years. Year should be incorporated as factor in the analysis to correctly assume the previous statements.

Regarding the plots used (4 replicates per each treatment), there are some inconsistencies along the manuscript. In tables where mean values of treatments and weeks per year are showed (tables 1-8), is indicated “Average of ten replications”. Whit this info, its seems that the sampling unit used as replicate for the analysis were each one of the 10 hills selected per plot. However, in the raw data shared, only 4 values appear (it seems that is one value per plot). So it is not clear what is the real sampling unit used as replicate for the analysis, plot or hills. Additionally, in the raw data shared, the BPH and predators abundances are denoted with decimals, indicating that maybe there is not the real raw data per replicate observed in the experiment, it seems that each value is the mean of the abundance observed in the 10 hills of each plot. Using a mean instead the real raw data observed, makes impossible to take into account the real variability of the variables invalidating all the results of the statistical analysis. For a correct analysis, data of each hill should be considered as a replicate and plots should be included in the experimental design as a random factor.

Another main issue found is related with the independence of the samples. As authors repeated the observation over the same hills along several weeks (or at least it is what seems from sentence of line 138), replicates are not independent along time. This mean that regular ANOVA could not be used, because one of the assumptions of this test is that the data must be independent. To include week as factor, they should perform a repeated measures ANOVA or a mixed ANOVA. Additionally, the other two assumptions of ANOVA (the responses for each factor level have a normal population distribution, and these distributions have the same variance) were not checked by the authors. This invalidates again the results obtained in this study. Before perform ANOVA, repeated measures ANOVA or mixed ANOVA the corresponding assumptions should be verified.

Moreover, analysis and results around BPH peak population data showed in figures 2 and 4 are not clear, because no info regarding the estimation of this variable and the analysis performed were included in the material and methods section.

Regarding Mirid bug population y Rove beetle population analysis, it sems from tables 6-8 that no individuals were observed in some weeks. However, as the variable was measured looking for individuals of these species during all the weeks, 0 values should be included in the statistical analysis. It is not the same “Na” than 0 values.

On the other hand, the olfactory response studies were not explained clearly in material and methods sections. Authors used as variable the percent attraction towards the source, for this estimation they need the data of the 20 spiders used in the bioassay. How they perform a t-test if they do not have replicates of percent values? Were always the same 20 spiders across the different plats studied? They do not explain anywhere the different sources tested. I suggest authors use a Chi-square test as used in other similar studies. Additionally, raw data of this study was not included in the manuscript. Authors should describe this analysis with sufficient information to be reproducible.

In addition to the variables tested, I suggest to authors analyze if there is any correlation among the abundance of BPH and their predators. These results could show some interesting patterns.

Validity of the findings

Following the different issues found around the statistical analysis, the data are not robust, statistically sound, and controlled. Additionally, some of the results were wrongly interpreted by authors. For example, the significant interaction among treatments and week observed in spiders 2020 (table 4) and Mirid bug in 2019 (table 5) were not considered by authors, thereby not identifying the changes in significant differences among treatment along the weeks. There is another example in line 246, in where authors stated that “rove beetle population did not differ significantly across the treatments and weeks in both the kharif seasons”, but in table 7 factor week seems to be significant. Finally, even authors commented in line 254 that “Rice grain yield differed significantly between the treatments” there is not statistical results about this variable along all the manuscript.

Additional comments

Some other minor comment should be considered:
-BPH should be defined at the beginning of the main text of the manuscript and not only in the abstract.
-line 112 “Transplanting was done on 22nd July and 30th July in2019 and 2020, respectively”. Write directly both dates without using “respectively”
-line 178: here is the first time that “kharif” is mentioned. This not English word should be explained previously; for example in material and methods section.

---

## Round 0.2 · Minor Revisions

Dear Authors,
I am sharing with you the .pdf file will my comments. The Journal will mail you the Word corresponding file for your convenience in executing the revisions.
Thank you very much.
Your Academic Editor

---

## Round 0.3 · Minor Revisions

Dear Author,

The Section Editor asks you fix minor issues:

1) add the post-hoc statistical analysis (Different letters indicating different means) to the figures. Letters are in the corresponding tables, but casual readers will look at the figures.

2) split each figure in an "A" and "B” per year data.

3) the text needs professional English editing, i.e. line 15 "plays like" should be "act as”, line 29 "Rice yield enhanced in" should be "Rice yield was enhanced in”, and line 39 "that provides nutrition" should be "and provide nutrition”. There is a need for a more formal text.

---

## Round 0.4 · Minor Revisions

Dear Authors, thank you for the changes to the figures and posthoc letters. The English language still needs to be improved, please consider that if you do not improve your English and provide a certificate of editing on this round I will Reject the paper.

---

## Round 0.5 · accepted · Accept

Dear authors, addressing all the editorial requests allows me to accept your submission for publication on PeerJ. The English still deserves some attention, let me suggest using the facilities the journal offers.

Thank you very much,
Francesco